# Research on Active Obstacle Avoidance of Intelligent Vehicles Based on Improved Artificial Potential Field Method

**Jing Tian** [1] , **Shaoyi Bei** [2,*], **Bo Li** [2,*] , **Hongzhen Hu** [2] , **Zhenqiang Quan** [2], **Dan Zhou** [2], **Xinye Zhou** [2] and **Haoran Tang** [2]

1  College of Mechanical Engineering, Jiangsu University of Technology, Changzhou 213001, China; tj_0506@163.com
2  College of Automobile and Traffic Engineering, Jiangsu University of Technology, Changzhou 213001, China; huhongzhen7171@163.com (H.H.); quanzhenqiang@163.com (Z.Q.); zd13952018530@163.com (D.Z.); e2090391825@163.com (X.Z.); gelare594054103@163.com (H.T.)
*  Correspondence: bsy1968@126.com (S.B.); bolifly311@gmail.com (B.L.)

**Abstract:** In the study of autonomous obstacle avoidance of intelligent vehicles, the traditional artificial potential field method has the problem that the vehicle may fall into the local minima and lead to obstacle avoidance failure. Therefore, this paper improves the traditional potential field function. Based on the vehicle dynamics model, a strategy of jumping out of local minima based on smaller steering angles is proposed. By finding a smaller steering angle and setting a suitable jump out step length, the intelligent vehicle is enabled to jump out of the local minima. Simulation experiments by MATLAB show that the improved method can jump out of the local minima. By comparing the planned trajectories of the traditional method and the improved method in static and dynamic obstacles situations, the trajectory planned by the improved method is smooth and the curvature is smaller. The planned trajectory is tracked by the Carsim platform, and the test results show that the improved method reduces the front steering wheel angle while the intelligent vehicle satisfies the vehicle dynamics constraints during active obstacle avoidance, which verifies the stability and rationality of the improved method.

**Keywords:** artificial potential field; local minima; path planning; intelligent vehicle

## 1. Introduction

Intelligent vehicles are advanced cars that integrate technologies such as computer science, sensors, and data processing. It has received extensive attention and research in academia and has been applied to some extent. Currently, most of the intelligent vehicles use advanced sensing technology to obtain vehicle location, speed, and other data through data fusion technology to achieve the extraction and analysis of characteristic data information between "human-vehicle-road", and finally making the vehicle environmentally aware, enabling autonomous decision-making and motion planning capabilities [1].

Path planning, as an important technology for intelligent vehicles to achieve autonomous driving, frequently refers to motions of a vehicle in a 2D or 3D world that contains obstacles [2]. The path planning technology of intelligent vehicles is to analyze and process the data collected by vehicle sensors and plan the vehicle trajectory independently without a human driver through a certain algorithm. This technology can not only plan the appropriate trajectory, but also reduce the traffic accidents caused by road environment changes and driver's operation errors, which is important to build a safe, efficient, and convenient driving environment.

According to the degree of grasp of environmental information, path planning can be divided into global path planning based on a priori complete information and local path planning based on sensor information [3,4]. The path planning techniques applied in automated driving scenarios can be roughly divided into four categories: graph search

based planners, sampling based planners, interpolating curve planners, and numerical optimization approaches [5]. Here are common path planning algorithms used in early applications for intelligent vehicles: (1) Grid method [6,7]: The grid method was earlier applied to the path planning of robots. It divides the robot's workspace into a grid, and the size of the grid cells is determined by the smallest rectangular space in which the robot can move freely, and the path is planned by calculating the shortest distance between the grids. (2) Dijkstra algorithm: Dijkstra's algorithm was proposed by the Dutch computer scientist Dijkstra in 1959 [8]. It is a classic graph searching algorithm; however, it has disadvantages such as high data computation and low efficiency. (3) A* algorithm [9,10]: It is an extension of Dijkstra's graph search algorithm. Its most important design aspect is the determination of the cost function, which defines the weights of the nodes [5]. (4) RRT (rapidly exploring random trees) algorithm [11,12]: RRT algorithm is a random sampling planning algorithm. It permits fast planning in semi-structured spaces by performing random searches in the navigation area [13].

The classical graph searching algorithms and sampling-based algorithm mentioned above have shortcomings and drawbacks. Vega-Brown et al. [14] proposed some asymptotically optimal algorithms for motion planning problems, the ideas in these algorithms can likely be combined with the ideas in many other sampling-based motion planning and graph search algorithms. Other scholars have also proposed some novel path planning algorithms based on advanced sensors and natural sciences: Nakrani et al. [15] designed a fuzzy-based obstacle avoidance navigation controller, which obtains information from an ultrasonic sensor array. Chen Y. et al. [16] proposed padding mean neural dynamic model (PMNDM), planning paths by transmitting nerve impulses in a topologically organized network. Jafari M. et al. [17] proposed a novel biologically inspired approach based on a computational model of emotional learning in mammalian limbic system, and it is applied for the first time in a synthetic UAV (Unmanned Aerial Vehicles) path planning scenario.

In real driving scenarios, when the drivers or sensors detect obstacles, they will slow down or steer to avoid the obstacles. This shows that the obstacle avoidance behavior of the vehicle depends on whether the location of the obstacles will influence the driving safety of the vehicle. Although the vehicle does not contact the obstacles during obstacle avoidance, the vehicle's motion is altered. This indicates that there is at least one virtual force acting on the vehicle, and this virtual force belongs to a field force [18]. This means that there is a potential field that changes the motion of the vehicle during the obstacle avoidance process. Therefore, we can explain the phenomenon of vehicle obstacle avoidance process in terms of fields and field forces.

The artificial potential field (APF) method has received much attention and research for its simplicity of calculation and high real-time performance. The APF method was first proposed by Khatib [19] in 1986 and it is commonly used for local path planning. The principle of APF method is to transform the real environment into a virtual potential field, which is the attractive field generated by the target and the repulsive field generated by the obstacle, and the vector superposition of the attractive field and repulsive field forms the combined potential field. Ultimately, the intelligent vehicle plans its path under the action of the combined potential field.

However, the traditional APF method has many drawbacks, the most typical of which is the problem of local minima and unable to reach target. To address the problem of local minima, scholars at home and abroad have proposed several improved methods. Choe, T.S. et al. [20] proposed the concept of steering potential fields, using overlapping integrated force fields to avoid obstacles and follow the planned path. Huang, Y. et al. [21] generate a diamond-shaped grid on a local map, add a voltage source between the starting point and the target, assign the resistance value of each edge on the grid by the APF method, and plan the local path by calculating the current maximum. Fan, S. [22] optimized the gravity model by setting tracking target points and optimized the potential field function by adjusting factors to eliminate local minima. Li, E. [23] proposed the SOPC-APF method

that introduces the idea of collision prediction and makes decisions before the robot enters the trap area or the minimum point problem.

APF method can also be combined with other intelligent algorithms. Bounini, F. et al. [24] find the actual path in the potential field according to a potential gradient descent algorithm and add a repulsive potential field to the current state with a local minimum. Cummings, M.L. [25] studied the path planning problem by combining the ant colony algorithm with the APF algorithm. Luo, D. et al. [26]. proposed to use the grid method to establish the planning space and use the ant colony algorithm as the global path search strategy to search for the optimal path. In addition, due to the early application of the APF method to robot kinematics, the restrictive constraints of the intelligent vehicle must be considered when it is referenced as a local path planning algorithm for intelligent vehicles.

To address the above problems, this paper first improves the potential field functions. Second, the road boundary potential field is added to constrain the motion region and limit the lateral motion of the intelligent vehicle. Finally, combining with the vehicle dynamics model, a strategy for jumping out of local minima based on smaller steering angles is proposed when the intelligent vehicle falls into local minima. The intelligent vehicle will search for potential jump-out points in the direction of smaller steering angles within the maximum steering angle range of the intelligent vehicle. The improved APF method is verified and analyzed by MATLAB and Carsim platform, and the improved APF method can successfully jump out of the local minima, at the same time, comparing the trajectory curvature before and after the improvement in the same obstacle environment. The trajectories planned by the improved APF method has smaller curvature and are easier to be tracked under the same Carsim tracking model. This paper provides a new idea for intelligent vehicle path planning based on potential field theory.

## 2. APF Method

### 2.1. Traditional APF Method

The principle of APF is to assume that the vehicle is in a virtual space; the attractive potential field generated by the target and the repulsive potential field generated by the obstacles. Attractive field has low potential energy and both start point and repulsive field have high potential energy. The superposition of these two potential fields forms an undulating space as shown in Figure 1 [27].

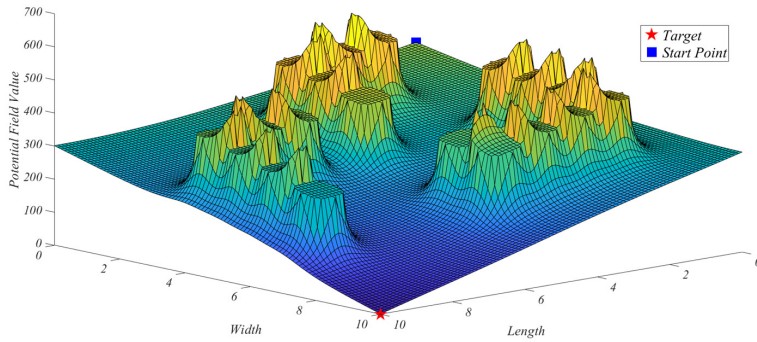

**Figure 1.** Potential fields.

In attractive field, the magnitude varies with the distance between the vehicle position and the target. The potential field function in this paper is referenced to Zhu, W. [28] and the attractive potential field is defined as:

$$U_{att}(X) = \frac{1}{2} K_a \rho^2 (P, P_g) \tag{1}$$

where $U_{att}(X)$ is the attractive potential field, $K_a$ is the attractive field coefficient, $\rho(P, P_g)$ is a vector whose magnitude is the Euclidean distance between the vehicle position $P$ and the target position $P_g$, the direction is the vehicle position toward the target position.

The attractive force is the negative gradient of the attractive potential field which can be calculated from the following equation:

$$F_{att}(X) = -\nabla U_{att}(X) = K_a \rho(P, P_g) \tag{2}$$

Repulsive fields are virtual potential fields generated by obstacles. Each obstacle has its own range of influence. When the vehicle is not within the influence range of an obstacle, the potential energy magnitude of the vehicle is zero; when the vehicle enters the influence range of the obstacle, the potential energy of the vehicle varies with the distance between the vehicle position and the obstacles. The repulsive potential field is defined as:

$$U_{rep}(X) = \begin{cases} \frac{1}{2}K_r\left(\frac{1}{\rho(P,P_{obs})} - \frac{1}{\rho_0}\right)^2 & 0 < \rho(P, P_{obs}) \leq \rho_0 \\ 0 & \rho(P, P_{obs}) > \rho_0 \end{cases} \tag{3}$$

where $U_{rep}(X)$ is the repulsive potential field, $K_r$ is the repulsive field coefficient, $\rho(P, P_{obs})$ is a vector whose magnitude is the Euclidean distance between the vehicle position $P$ and the obstacle position $P_{obs}$, the direction is from the obstacle to the vehicle, $\rho_0$ is the radius of the obstacle's influence range.

The repulsive force is the negative gradient of the repulsive potential field which can be calculated from the following equation:

$$F_{rep}(X) = -\nabla U_{rep}(X) = \begin{cases} K_r\left(\frac{1}{\rho(P,P_{obs})} - \frac{1}{\rho_0}\right)\frac{1}{\rho^2(P,P_{obs})} & 0 < \rho(P, P_{obs}) \leq \rho_0 \\ 0 & \rho(P, P_{obs}) > \rho_0 \end{cases} \tag{4}$$

Vehicle is often affected by the repulsive field of multiple obstacles while moving toward the target. Therefore, the combined force field is a superposition of an attractive field and multiple repulsive fields.

The combined force potential field function can be expressed as:

$$U_{total}(X) = U_{att}(X) + \sum_{i=1}^{n} U_{rep}(X) \tag{5}$$

In the above equation, $U_{total}(X)$ is the combined potential field and $n$ is the number of obstacles.

The combined force is the negative gradient of the force potential field and can be expressed as:

$$F_{total}(X) = -\nabla U_{total}(X) = F_{att}(X) + F_{rep}(X) \tag{6}$$

Ultimately, as shown in Figure 2, the combined forces control the motion of the vehicle.

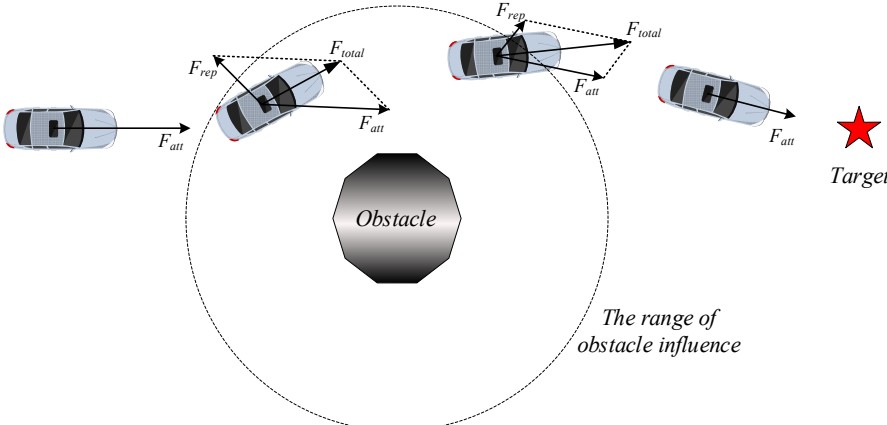

**Figure 2.** Forces on the vehicle.

*2.2. Shortcomings of Traditional APF Method*

1.  Because the attractive force is proportional to the distance between vehicle and target, an excessive attractive force will cause the vehicle to hit a nearby obstacle when the vehicle is far from the target.
2.  The attractive force generated by the target is zero when the vehicle reaches the target. Assuming there is an obstacle near the target, the vehicle will still be repelled away from the target by the repulsive force generated by the obstacle at this time, which causes the vehicle to oscillate near the target and cannot reach it;
3.  In practical urban driving scenarios, vehicles must move on a defined road and cannot move outside the road boundaries. Therefore, adding a road boundary potential field to limit the lateral motion of the vehicle is necessary;
4.  Assume that the vehicle is at a point where the combined force on the vehicle is zero. The vehicle will fall into a local minima if the vehicle does not reach the target. It is the most common fault of the traditional APF method.

## 3. Vehicle Dynamics Model

The vehicle dynamics model is very essential for designing feasible trajectory. The planned trajectory should be limited within the vehicle dynamics constraints [29], and the two-degree-of-freedom vehicle dynamics model established in this paper is shown in Figure 3. In this figure: $F_{yf}$ and $F_{yr}$ represent the lateral force of the front wheel and the rear wheel; $V_x$ and $V_y$ are the longitudinal speed and the lateral speed, respectively; $\varphi$ is the yaw angle; $\beta$ is the lateral deflection angle of the center of mass; $\alpha_f$ and $\alpha_r$, denote the slip angle of the front and rear axle, respectively; $l_f$ is the distance from the center of mass to the front axle; $l_r$ is the distance from the center of mass to the rear axle; $\delta$ represents the front tire angle.

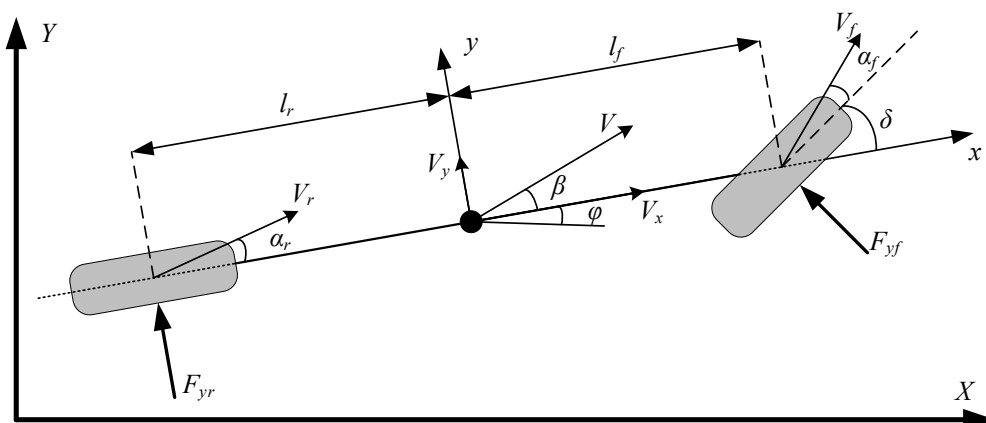

**Figure 3.** 2-DOF (degree-of-freedom) vehicle dynamics model.

Based on the balance of forces in the *y*-axis direction, the vehicle dynamics equilibrium equation can be expressed as:

$$F_{yf}\cos\delta + F_{yr} = ma_y \tag{7}$$

The relationship between lateral force of the front wheel and the rear wheel, lateral deflection angle and lateral deflection stiffness can be expressed as:

$$\begin{cases} F_{yf} = C_f\alpha_f \\ F_{yr} = C_r\alpha_r \end{cases} \tag{8}$$

where $C_f$ and $C_r$ are the front wheel lateral deflection stiffness and the rear wheel lateral deflection stiffness, respectively. According to the coordinate system, the lateral deflection angle of the front and rear wheels of the vehicle can be expressed as:

$$\begin{cases} \alpha_f = \dfrac{\dot{\varphi}l_f + V_y}{V_x} - \delta \\ \alpha_r = \dfrac{V_y - \dot{\varphi}l_r}{V_x} \end{cases} \tag{9}$$

Assuming the front tire angle $\delta$ at a small angle, we can regard $cos\delta \approx 1$. According to the Equations (8) and (9), Equation (7) can be transformed as:

$$m(\ddot{y} + V_x\dot{\varphi}) = C_f\left(\frac{\dot{\varphi}l_f + \dot{y}}{V_x} - \delta\right) + C_r\left(\frac{\dot{y} - \dot{\varphi}l_r}{V_x}\right) \tag{10}$$

Note that the velocity and acceleration in the *y*-axis direction can be expressed as:

$$\begin{cases} V_y = \dot{y} \\ a_y = \ddot{y} + V_x\dot{\varphi} \end{cases} \tag{11}$$

Finally, the vehicle dynamics model based on the front tire angle $\delta$ can be expressed as:

$$\ddot{y} = \frac{C_f + C_r}{mV_x}\dot{y} + \left(\frac{l_fC_f - l_rC_r}{mV_x} - V_x\right)\dot{\varphi} - \frac{C_f}{m}\delta \tag{12}$$

## 4. Improved APF Method

Based on the previous analysis of the traditional APF method, the details of the improved APF method are as follows:

### 4.1. Potential Field Functions

To avoid the problem of excessive initial attractive force of the vehicle at the starting point, the attractive field function could be modified. The improved attractive potential field function can be expressed as:

$$U_{att}(X) = \begin{cases} \varepsilon K_a\rho(P, P_g) & \rho(P, P_g) \geq d_0 \\ \frac{1}{2}K_a\rho^2(P, P_g) & \rho(P, P_g) < d_0 \end{cases} \tag{13}$$

The corresponding attractive force can be expressed as:

$$F_{att}(X) = \begin{cases} -\varepsilon K_a\dfrac{\rho(P, P_g)}{|P - P_g|} & \rho(P, P_g) \geq d_0 \\ -K_a\rho(P, P_g) & \rho(P, P_g) < d_0 \end{cases} \tag{14}$$

where $d_0$ is the threshold value for the distance between the vehicle and the target; $\varepsilon$ is the attractive field modulation factor. The most significant difference after the improvement is that when the distance between the vehicle and the target is greater than $d_0$, the gravitational force is considered constant.

To address the problem of vehicle oscillation near the target, an adjustment factor $\rho^n(P, P_g)$ is added to the repulsive field function. The improved repulsive field function can be expressed as:

$$U_{rep}(X) = \begin{cases} \frac{1}{2}K_r\left(\frac{1}{\rho(P, P_{obs})} - \frac{1}{\rho_0}\right)^2\rho^n(P, P_g) & 0 < \rho(P, P_{obs}) \leq \rho_0 \\ 0 & \rho(P, P_{obs}) > \rho_0 \end{cases} \tag{15}$$

The corresponding repulsive force can be expressed as:

$$F_{rep}(X) = \begin{cases} \vec{n}_{o,v}F_{rep1} + \vec{n}_{v,g}F_{rep2} & \rho(P, P_{obs}) \leq \rho_0 \\ 0 & \rho(P, P_{obs}) > \rho_0 \end{cases} \tag{16}$$

where the direction of repulsive force $F_{rep1}$ is from the obstacle toward the vehicle, and the direction of repulsive force $F_{rep2}$ is from the vehicle toward the target. The magnitude of $F_{rep1}$ and $F_{rep2}$ is determined by the following equation:

$$
\begin{cases}
F_{rep1} = K_r \left( \dfrac{1}{\rho(P,P_{obs})} - \dfrac{1}{\rho_0} \right) \dfrac{\rho^n(P,P_g)}{\rho^2(P,P_{obs})} \\
F_{rep2} = \dfrac{n}{2} K_r \left( \dfrac{1}{\rho(P,P_{obs})} - \dfrac{1}{\rho_0} \right)^2 \rho^{n-1}(P,P_g)
\end{cases}
\tag{17}
$$

where $n$ is an arbitrary non-zero constant, taken as $n = 2$ in this paper.

The road boundary potential field is mainly used to restrict the driving area of intelligent vehicles, which is essentially a type of repulsive force. In this paper, taking a two-lane road as an example, the two green zones represent the area where the intelligent vehicle should be in motion as shown in the Figure 4. The road boundary potential field function is expressed by segmentation function, which can make vehicles move steadily within the road boundary. The road boundary potential field function can be expressed as:

$$
U_{road}(P) =
\begin{cases}
\dfrac{1}{3} K_{road} \left( P_y + \dfrac{D}{2} \right)^3 - D + \dfrac{W}{2} \le P_y < -\dfrac{D}{2} \\
-\dfrac{\lambda}{3} K_{road} \left( P_y + \dfrac{D}{2} \right)^3 - \dfrac{D}{2} \le P_y < 0 \\
\dfrac{\lambda}{3} K_{road} \left( P_y - \dfrac{D}{2} \right)^3 \ 0 < P_y \le \dfrac{D}{2} \\
-\dfrac{1}{3} K_{road} \left( P_y - \dfrac{D}{2} \right)^3 \ \dfrac{D}{2} < P_y \le D - \dfrac{W}{2}
\end{cases}
\tag{18}
$$

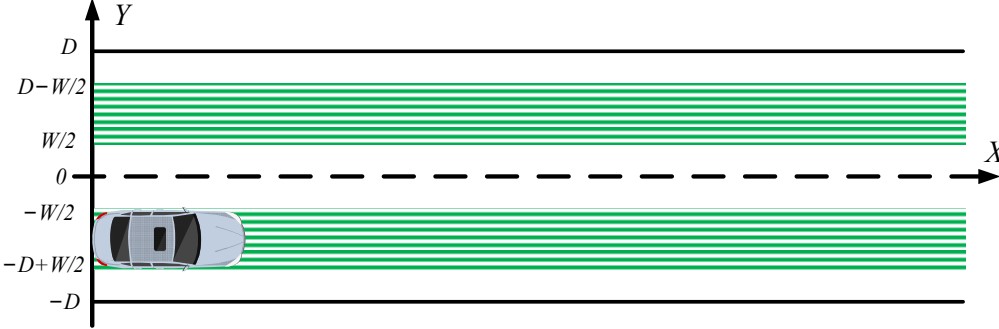

**Figure 4.** Intelligent vehicle driving areas.

The corresponding road boundary repulsive force can be expressed as:

$$
F_{road}(P) = -\nabla U_{road}(P) =
\begin{cases}
K_{road} \left( P_y + \dfrac{D}{2} \right)^2 - D + \dfrac{W}{2} \le P_y < -\dfrac{D}{2} \\
-\lambda K_{road} \left( P_y + \dfrac{D}{2} \right)^2 - \dfrac{D}{2} \le P_y < 0 \\
\lambda K_{road} \left( P_y - \dfrac{D}{2} \right)^2 \ 0 < P_y \le \dfrac{D}{2} \\
-K_{road} \left( P_y - \dfrac{D}{2} \right)^2 \ \dfrac{D}{2} < P_y \le D - \dfrac{W}{2}
\end{cases}
\tag{19}
$$

where $W$ is the vehicle width, $D$ is the lane width, $K_{road}$ is the road boundary potential field coefficient, $\lambda$ is the modulation factor, and $P_y$ is the vertical coordinate of the vehicle center of mass position in the road coordinate system $XOY$. The direction of repulsion of the road boundary potential field is perpendicular to the road boundary. When the intelligent vehicle can avoid the obstacle without changing lanes, the intelligent vehicle is restricted to remain in the current lane, unless the longitudinal repulsive force on the vehicle is greater than the maximum repulsive force of the road boundary without changing lanes to reduce the driving risk of changing lanes.

By superposition of the above three potential fields, the equilibrium equation for the vehicle subjected to the potential field can be obtained:

$$U_{total} = U_{att} + U_{rep} + U_{road} \tag{20}$$

The equilibrium equation for the forces on the vehicle can be expressed as:

$$F_{total} = F_{att} + F_{rep} + F_{road} \tag{21}$$

### 4.2. Strategies for Jump out of Local Minima Based on Smaller Steering Angles

The intelligent vehicle will stall or oscillate when it falls into local minima. To solve this problem, this paper proposed a strategy, that is searching for potential jump out points within the steering angle range as shown in Figure 5. The specific steps are as follows:

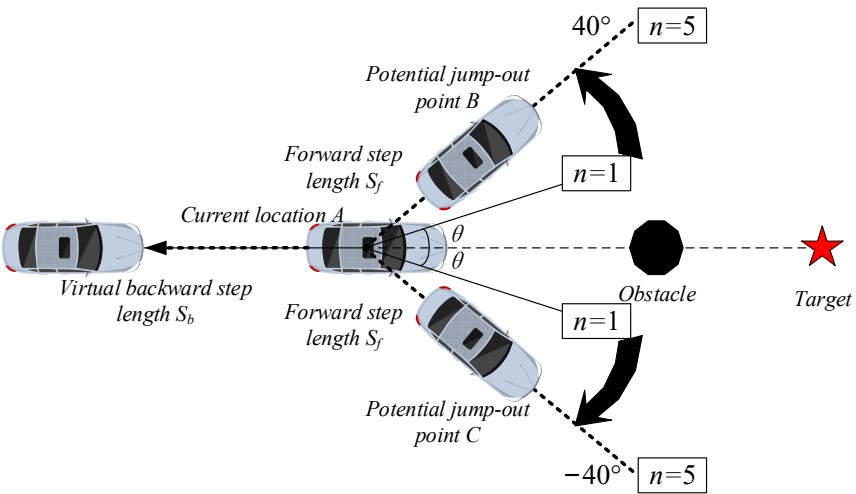

**Figure 5.** Strategy for jumping out of local minima.

When the vehicle has not reached the target, the vehicle satisfies the equation:

$$\left| P - P_g \right| \geq step\ length \tag{22}$$

The local minima are defined as:

$$\left| P_{i+2} - P_i \right| \leq 0.1 * step\ length\ ||\ F_{sum} = 0 \tag{23}$$

where *i* is the serial number of step. When the intelligent vehicle falls into the local minima, the coordinates of the intelligent vehicle at that moment are noted as $A(x_0, y_0)$, the direction of vehicle motion at that moment is taken as the reference line. Initializing the number of angle changes $n = 1$, and picking the angles on both sides of the reference line direction, the angles $\theta_{th}$ are determined by Equations (24) and (25):

$$n_{th} = \pm \begin{bmatrix} \frac{1}{1024} \\ \frac{1}{256} \\ \frac{1}{64} \\ \frac{1}{16} \\ \frac{1}{4} \\ 1 \end{bmatrix} \begin{bmatrix} 1 & \cdots & n \end{bmatrix} \tag{24}$$

$$\theta_{th} = \frac{n_{th}}{|n_{th}|} * \sqrt{|320 * n_{th}|} \tag{25}$$

The magnitude of the front tire angle $\delta$ of a vehicle can be expressed as:

$$\delta = 2\pi * \frac{\theta_{th}}{360} + arctan\frac{F_{sumy}}{F_{sumx}} + \pi * (F_{sumx} < 0) \tag{26}$$

At the same time, according to Equation (27) to obtain the former step length $S_f$.

$$S_f = \begin{cases} 0.5 * step\ length\ \Delta U(X_{A_i} - X_{A_{i-2}}) \leq 0.8 \\ step\ length\ 0.8 < \Delta U(X_{A_i} - X_{A_{i-2}}) < 1.2 \\ 1.5 * step\ length\ \Delta U(X_{A_i} - X_{A_{i-2}}) \geq 1.2 \end{cases} \tag{27}$$

where $\Delta U(X_{A_i} - X_{A_{i-2}})$ is the potential field magnitude difference between the point $A$ and the position twice the length steps before the point $A$. Based on the front tire angle $\delta$ and the former step length $S_f$, the vehicle gets two potential jump-out points, sit marked as $B(x_1, y_1)$ and $C(x_2, y_2)$.

Calculate the value of the potential field when the vehicle is at positions $A$, $B$, and $C$ respectively and write $U(x_0)$, $U(x_1)$, and $U(x_2)$. Comparing the magnitude of $U(x_0)$, $U(x_1)$ and $U(x_2)$:

If the potential field value satisfies both $U(x_0) < U(x_1)$ and $U(x_0) < U(x_2)$, it means that the potential field value at point $A$ is the lowest. Neither point $B$ nor point $C$ can be the suitable jump-out point. Then increase the number of angle changes once and perform the above step operation again. Meanwhile, the steering angle of the intelligent vehicle should satisfy its dynamics constraints in the actual driving scenario. The maximum steering angle of the vehicle does not exceed $40°$ and $n$ should be no more than 5. If $n$ is greater than 5, then the vehicle is set back two times the virtual step length to obtain a critical local minima, and performed again from the beginning.

If it does not satisfy $U(x_0) < U(x_1)$ or $U(x_0) < U(x_2)$, it means that point $B$ and point $C$ have at least one point with a lower potential field value than point $A$. Then comparing the magnitude of $U(x_1)$ and $U(x_2)$, the vehicle moves to point $C$ if potential field value satisfied $U(x_1) > U(x_2)$ and moves to point $B$ if potential field value satisfied $U(x_2) > U(x_1)$.

When determining the potential jump-out points, attention should be paid to whether the line between the current position of the intelligent vehicle and the potential jump-out point will collide with the obstacle. If a collision occurs, the potential jump-out point should be discarded and searched again. The planned trajectory and front tire angle should satisfy Equation (12). The flowchart of the strategy is shown in Figure 6.

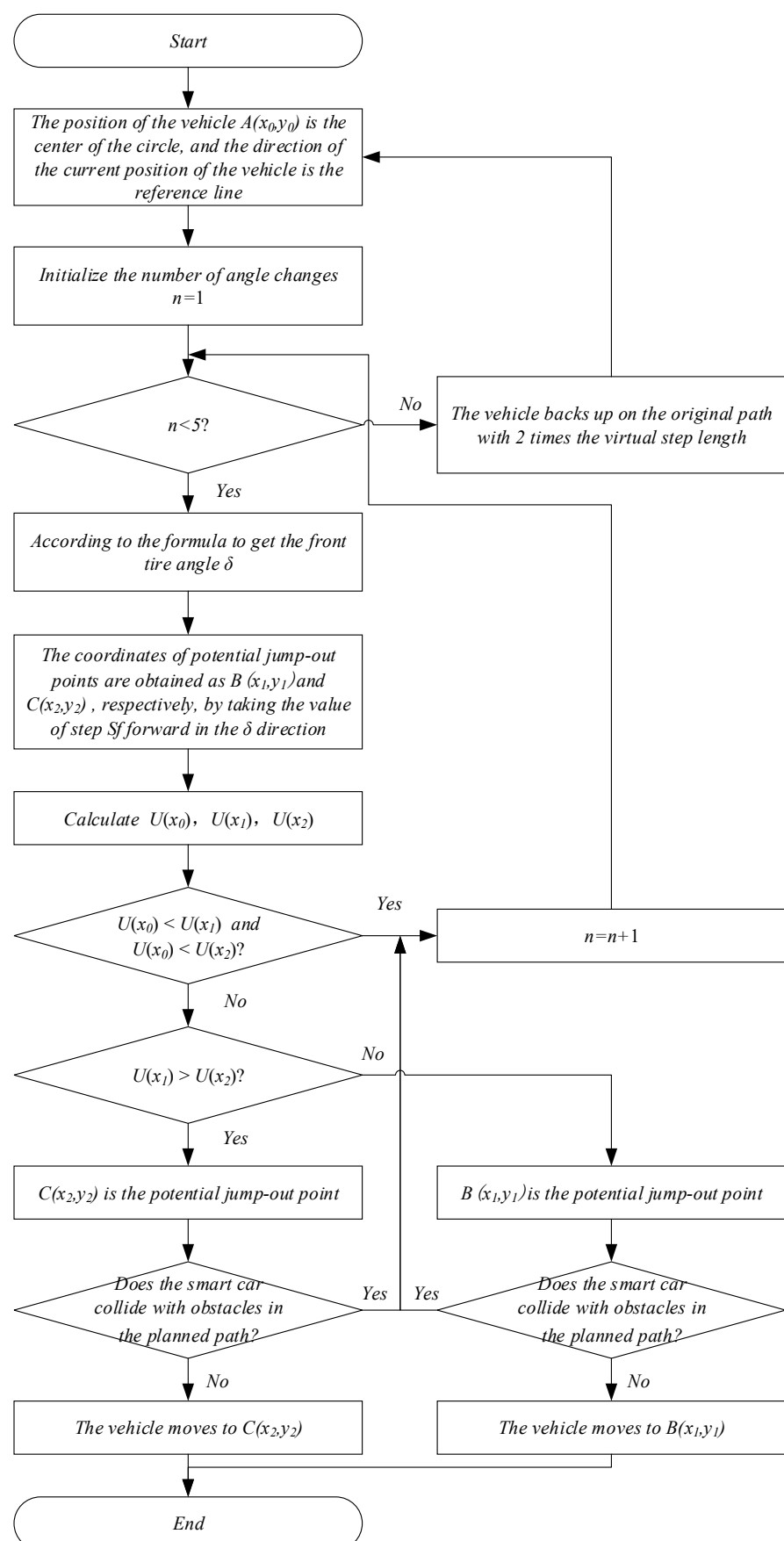

**Figure 6.** Flowchart of the strategy for jump out of the local minima.

## 5. Simulation and Analysis

### 5.1. Simulation Environment Construction

To verify the feasibility of the improved APF method in local path planning of intelligent vehicles, simulations are carried out on the MATLAB 2021a platform. The traditional and improved APF methods are compared to deal with the problem of local minima; the planning trajectories of the traditional and improved APF methods are compared in the static obstacles environment and the dynamic obstacles environment. The effectiveness of the improved method is evaluated by the curvature of the trajectory. The main parameters of the simulation experiments are shown in Table 1.

**Table 1.** Simulation parameters setting table.

| Parameter Name and Symbol Representation | Value/Unit |
|---|---|
| Attractive field action coefficient $K_a$ | 15 |
| Repulsive field action coefficient $K_r$ | 10 |
| Road boundary potential field coefficient $K_{road}$ | 20 |
| Radius of the influence range of the obstacle $\rho_0$ | 5 m |
| Vehicle length $L$ | 4.7 m |
| Vehicle width $W$ | 1.8 m |
| Lane width $D$ | 3.5 m |
| Step length | 0.1 m |

### 5.2. Analysis of Simulation Results

#### 5.2.1. Simulation Analysis of Local Minima Problems

Set the starting position of the main vehicle as (0 m, 0 m) and the target position as (50 m, 0 m). The obstacle coordinates are (25 m, 0 m) in the single obstacle case and (25 m, 3.5 m) (25 m, −3.5 m) in the multiple obstacles case. The simulation results of the traditional APF method are shown in Figure 7.

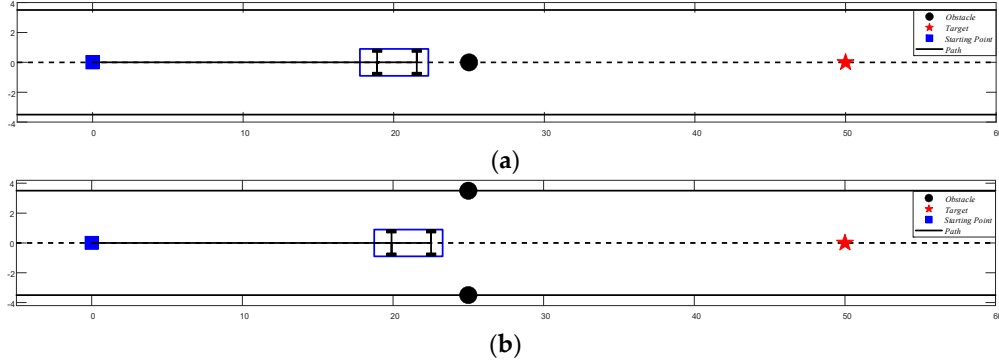

**Figure 7.** Local minima formed by traditional APF method. (**a**) Single obstacle; (**b**) multiple obstacles.

From Figure 7, it can be seen that the vehicles all fall into local minima, and the coordinates of the local minima are (19 m, 0 m) and (19.8 m, 0 m). The results of path planning are failures because they fail to achieve their targets. The simulation results of the improved APF method are shown in Figure 8.

From Figure 8, it can be seen that the vehicle jump out of the local minima by using the improved APF method, the planned trajectory is continuous and smooth. As can be seen from Figure 9, the absolute values of the curvature of the trajectories planned by the improved APF method are all less than 0.4 m$^{-1}$, which satisfies the requirements of path planning. It also indicates that the strategy of jump out of the local minima based on smaller steering angle is effective.

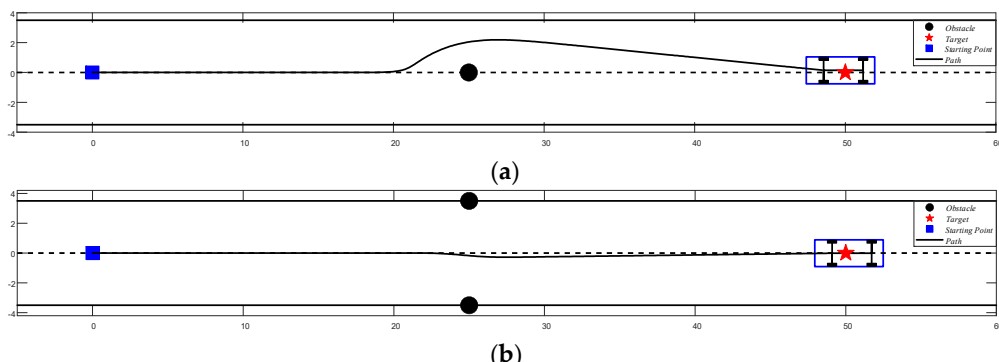

**Figure 8.** Trajectory planned by the improved APF method. (**a**) Single obstacle; (**b**) multiple obstacles.

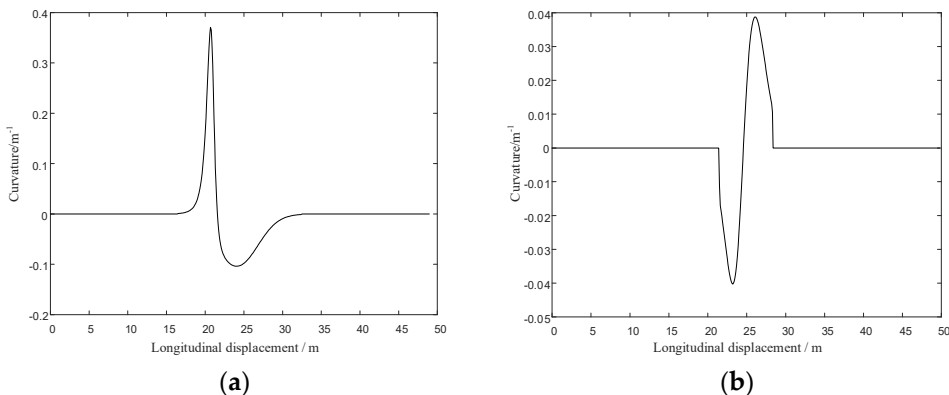

**Figure 9.** Curvature of the trajectory planned by the improved APF method; (**a**) trajectory curvature for single obstacle case; (**b**) trajectory curvature for multiple obstacles case.

### 5.2.2. Simulation of Path Planning in Static Obstacles Situations

Taking two lanes in both directions as an example, in the initial state, the position of the subject vehicle is (0 m, −1.75 m) and the target position is (100 m, 1.75 m). Assuming that the subject vehicle moves at a constant speed of 10 m/s, the initial positions of static obstacle vehicle 1 and static obstacle vehicle 2 are (15 m, 1.75 m) and (50 m, −2.5 m), respectively, the location relationships are shown in Figure 10.

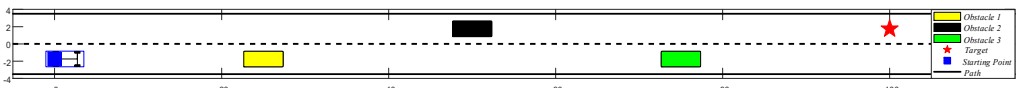

**Figure 10.** Static obstacles driving environment.

The simulation results of trajectory planned by the traditional and improved APF methods are shown in Section 5.2.2 and Figure 12.

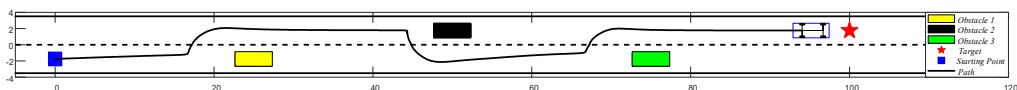

**Figure 11.** Trajectory planned by the traditional APF method in static obstacles driving environment.

Although the traditional APF method also successfully planned the trajectory, it can be seen that the improved APF method planned a smoother trajectory than the traditional APF method by comparing Section 5.2.2 and Figure 12, which is also confirmed by the curvature comparison in Figure 13.

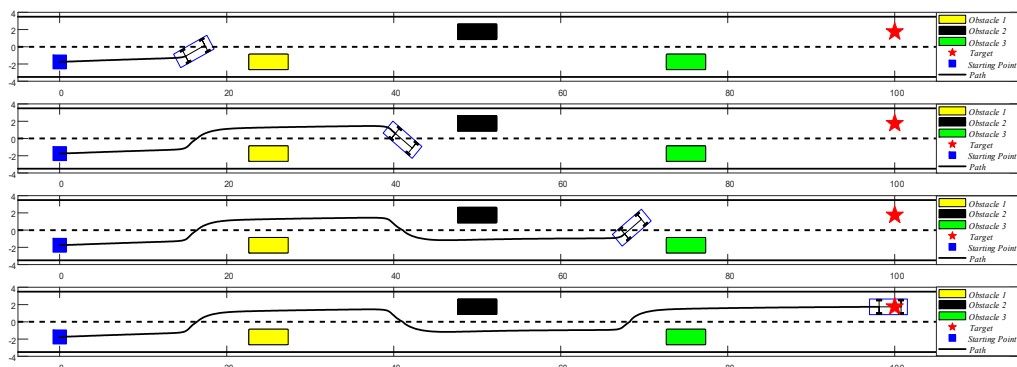

**Figure 12.** Trajectory planned by the improved APF method in static obstacles driving environment.

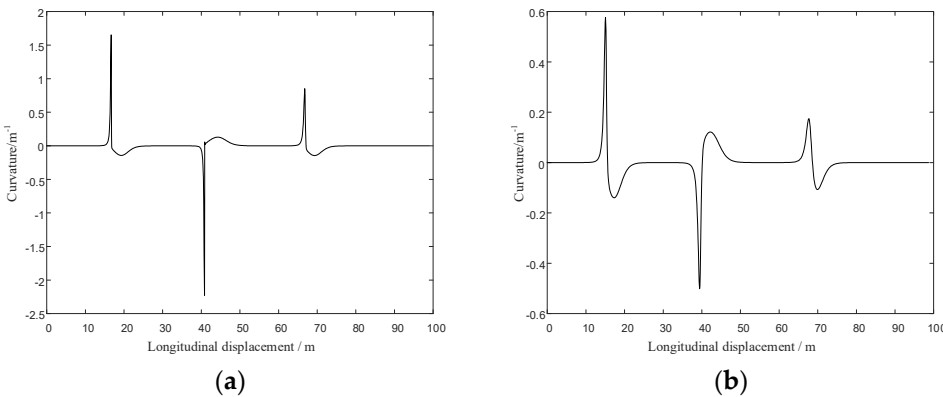

**Figure 13.** Comparison of curvature of planned trajectories in static obstacles driving environment; (**a**) curvature of planned trajectory by traditional APF; (**b**) curvature of planned trajectory by improved APF.

5.2.3. Simulation of Path Planning in Dynamic Obstacle Situation

Still take two lanes in both directions as an example, in the initial state, the initial position of the subject vehicle is (0 m, −1.75 m) and the target position is (100 m, 1.75 m). Assuming that the subject vehicle is moving at a constant speed of 10 m/s, the initial positions of dynamic obstacle vehicle 1 and dynamic obstacle vehicle 2 are (15 m, 1.75 m) and (50 m, −2.5 m) respectively, the location relationships are shown in Figure 14.

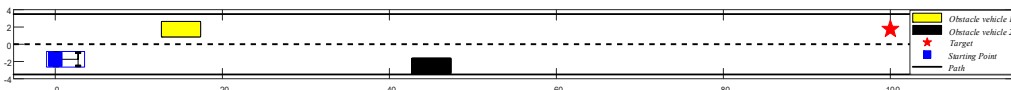

**Figure 14.** Dynamic obstacles driving environment.

Assume that dynamic obstacle vehicle 1 and dynamic obstacle vehicle 2 are at a uniform speed of 5 m/s and 3 m/s, respectively. The simulation results of trajectory planned by the traditional and improved APF methods are shown in Figures 15 and 16.

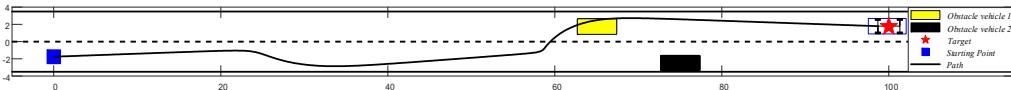

**Figure 15.** Trajectory planned by the traditional APF method in dynamic obstacles driving environment.

It can be seen from the figure that the trajectories are successfully planned by the APF method before and after the improvement. However, as can be seen from Figure 17, the peak curvature of the trajectory planned by the improved APF method is approximately half of the curvature of the trajectory planned by the traditional APF method. This means

that the improved APF method planned the trajectory more smoothly and the vehicle is easier to maneuver when avoiding obstacles.

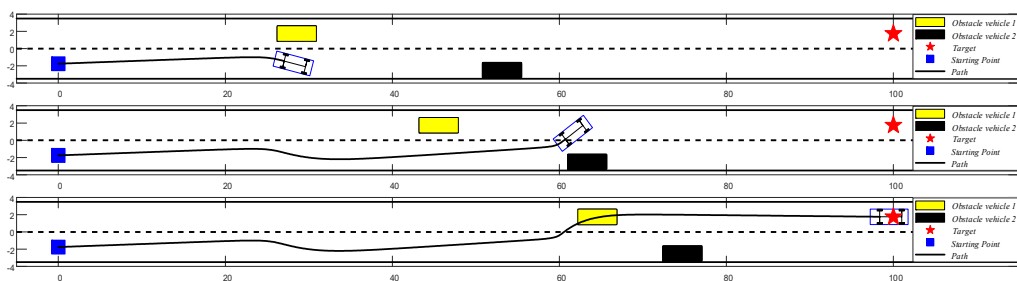

**Figure 16.** Trajectory planned by the improved APF method in dynamic obstacles driving environment.

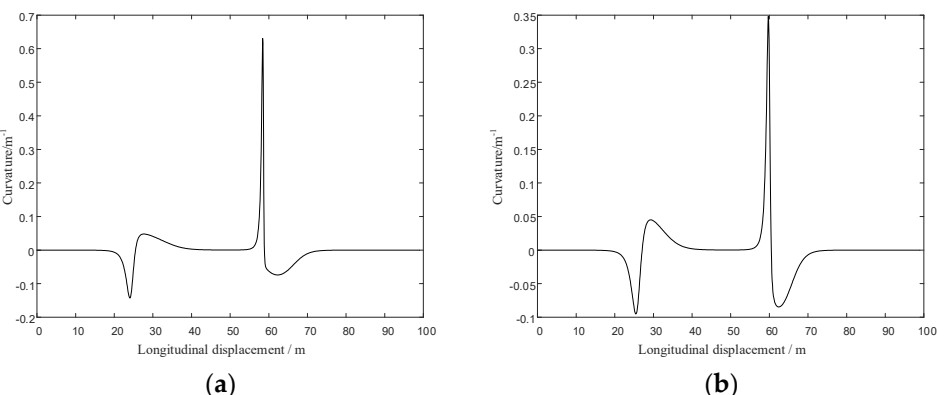

**Figure 17.** Comparison of curvature of planned trajectories in dynamic obstacles driving environment. (**a**) Curvature of planned trajectory by traditional APF; (**b**) curvature of planned trajectory by improved APF.

*5.3. Simulation of Tracking the Planned Trajectory*

In order to further verify the reliability of the path planning of the improved APF method, we have tracked the planned trajectories in static and dynamic obstacle driving environments on the Carsim simulation platform and analyzed the simulation results.

In order to exhibit the amount of steering wheel angle variation of the intelligent vehicle under different speeds, we set two groups of speed commonly used in urban road environment, 10 m/s and 15 m/s. These two groups of speed will be used as the expected speed of the subject vehicle to track the trajectory planned in the previous section. Since it is only necessary to obtain the variation rules of steering wheel angle before and after the improved APF method, the article only takes the default tracking model in Carsim. In static obstacle environment, Figure 18 compared the planned trajectory and the actual trajectory for expected vehicle speed of 10 m/s. The Euclidean errors of the planned trajectory and the actual trajectory for expected vehicle speed of 10 m/s are shown in Figure 19. The variation curves of steering wheel angle for expected vehicle speed of 10 m/s in static obstacles driving environment are shown in Figure 20.

Figure 21 compared the planned trajectory and the actual trajectory for expected vehicle speed of 15 m/s. The Euclidean errors of the planned trajectory and the actual trajectory for expected vehicle speed of 15 m/s are shown in Figure 22. The variation curves of steering wheel angle for expected vehicle speed of 15 m/s in static obstacles driving environment are shown in Figure 23.

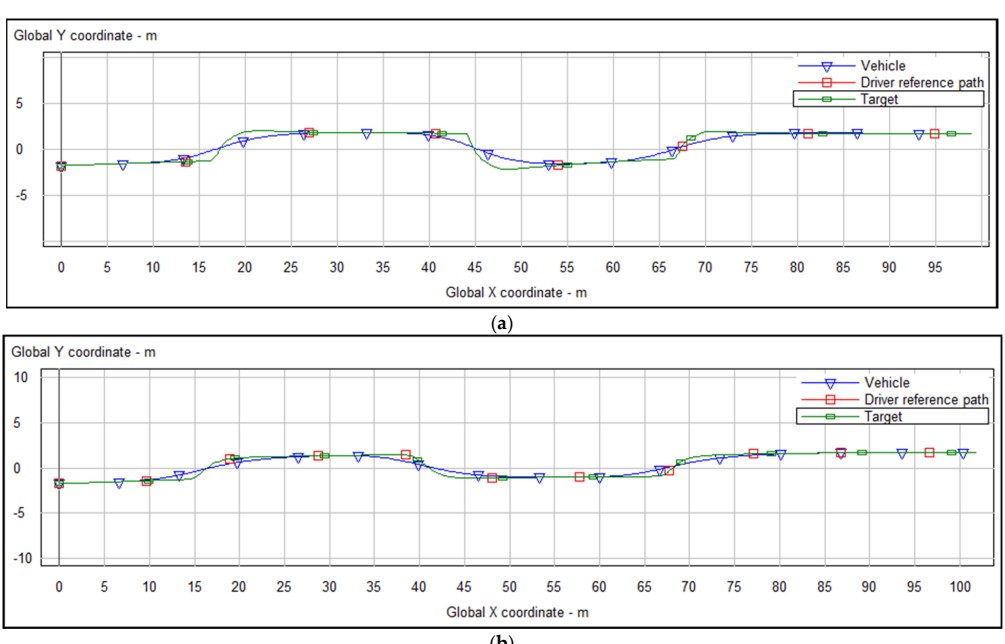

**Figure 18.** Comparison of planned trajectory and actual trajectory for expected vehicle speed of 10 m/s in static obstacles driving environment. (**a**) Planned trajectory by traditional APF and actual trajectory of vehicle; (**b**) planned trajectory by improved APF and actual trajectory of vehicle.

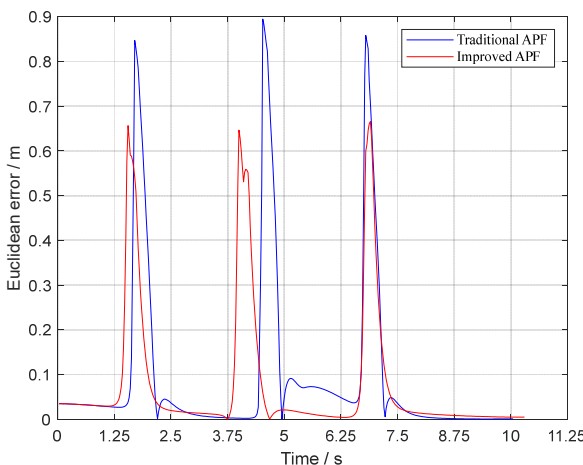

**Figure 19.** Euclidean errors of trajectories planned for expected vehicle speed of 10 m/s in static obstacles driving environment.

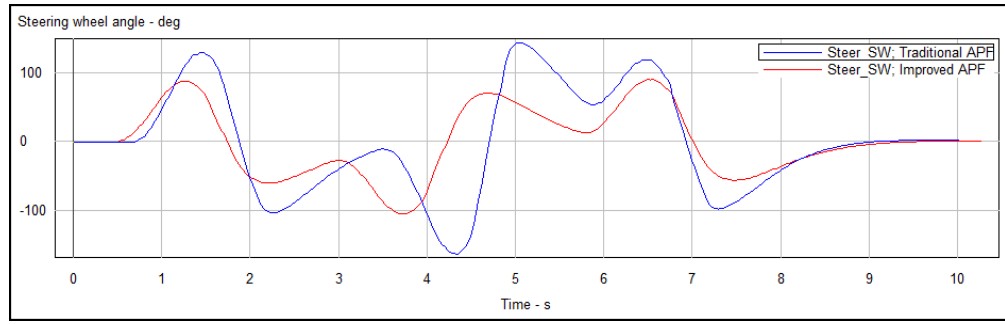

**Figure 20.** Steering wheel angles for expected vehicle speed of 10 m/s in static obstacles driving environment.

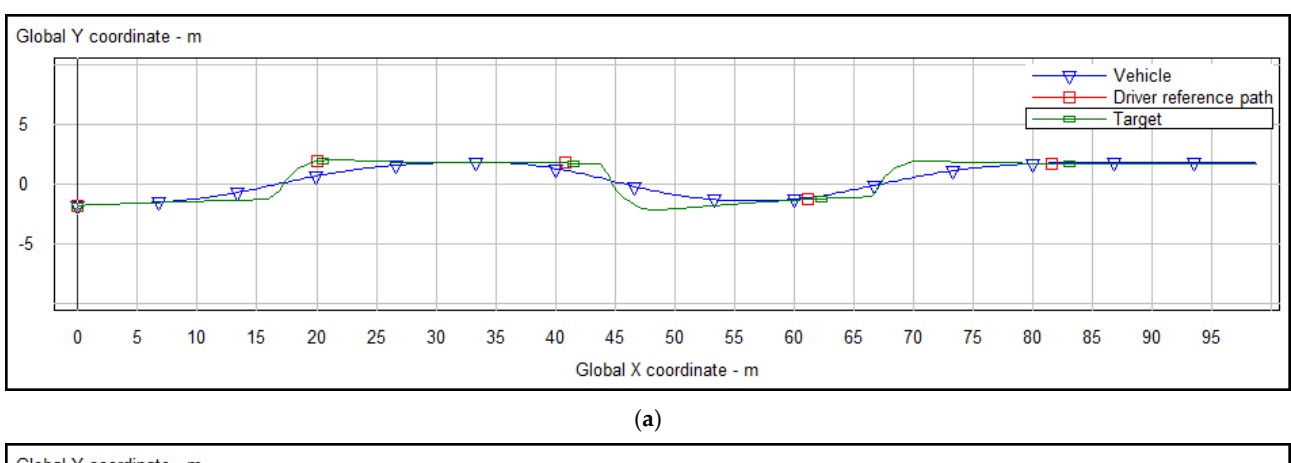

(**a**)

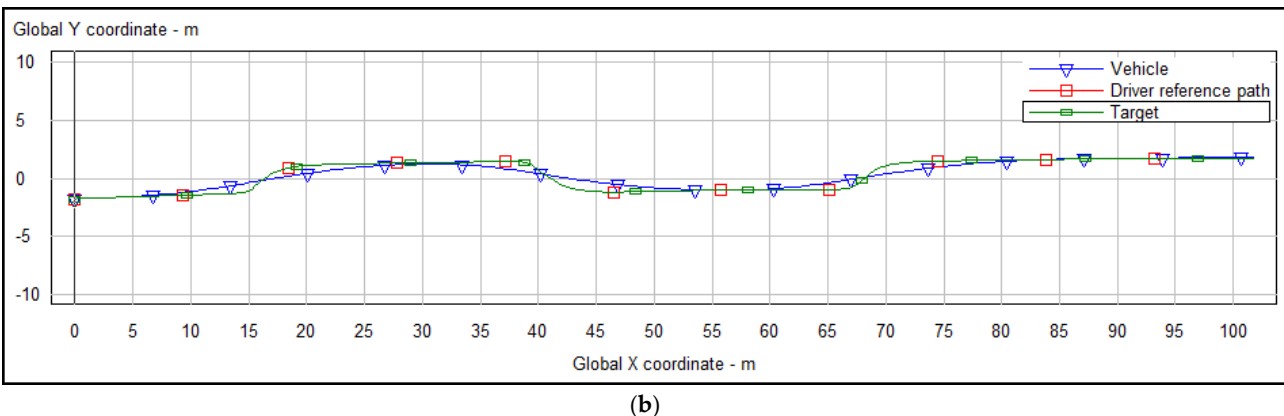

(**b**)

**Figure 21.** Comparison of planned trajectory and actual trajectory for expected vehicle speed of 15 m/s in static obstacles driving environment. (**a**) Planned trajectory by traditional APF and actual trajectory of vehicle; (**b**) planned trajectory by improved APF and actual trajectory of vehicle.

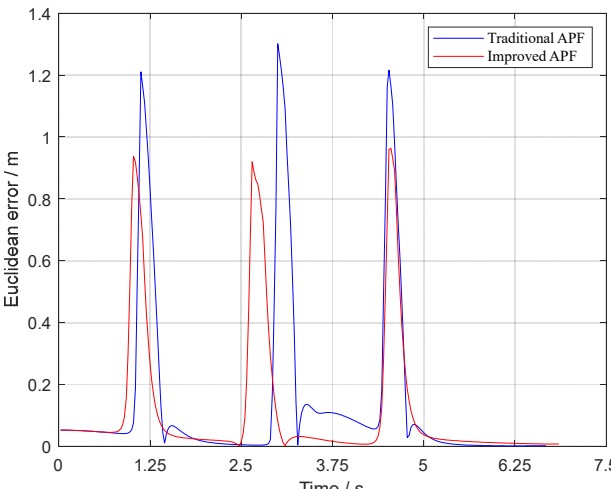

**Figure 22.** Euclidean errors of trajectories planned for expected vehicle speed of 15 m/s in static obstacles driving environment.

In dynamic obstacles environment, Figure 24 compared the planned trajectory and the actual trajectory for expected vehicle speed of 10 m/s. The Euclidean errors of the planned trajectory and the actual trajectory for expected vehicle speed of 10 m/s are shown in Figure 25. The variation curves of steering wheel angle for expected vehicle speed of 10 m/s in dynamic obstacles driving environment are shown in Figure 26.

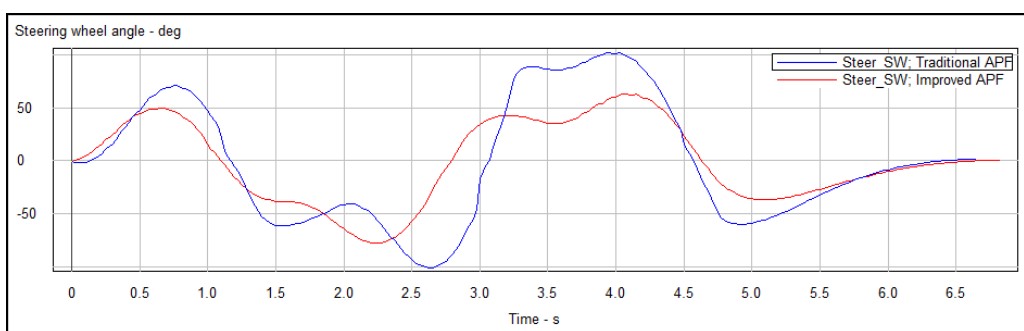

**Figure 23.** Steering wheel angles for expected vehicle speed of 15 m/s in static obstacles driving environment.

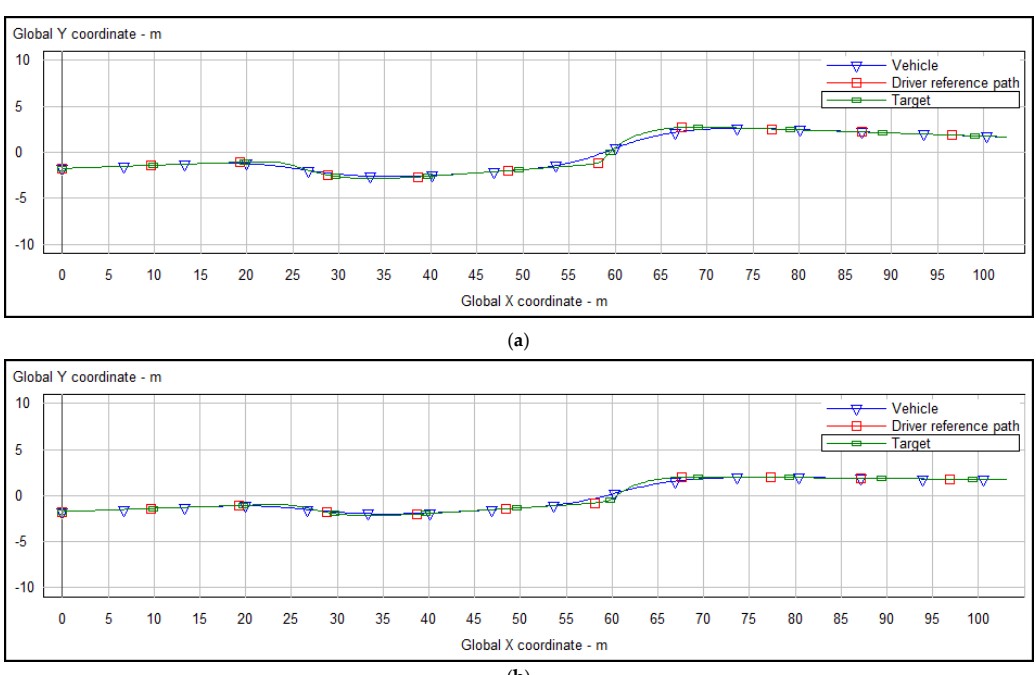

**Figure 24.** Comparison of planned trajectory and actual trajectory for expected vehicle speed of 10 m/s in dynamic obstacles driving environment. (**a**) Planned trajectory by traditional APF and actual trajectory of vehicle; (**b**) planned trajectory by improved APF and actual trajectory of vehicle.

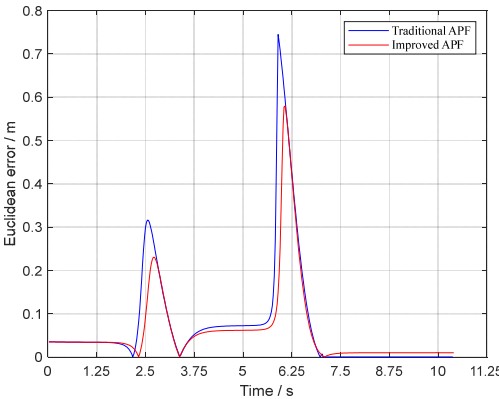

**Figure 25.** Euclidean errors of trajectories planned for expected vehicle speed of 10 m/s in dynamic obstacles driving environment.

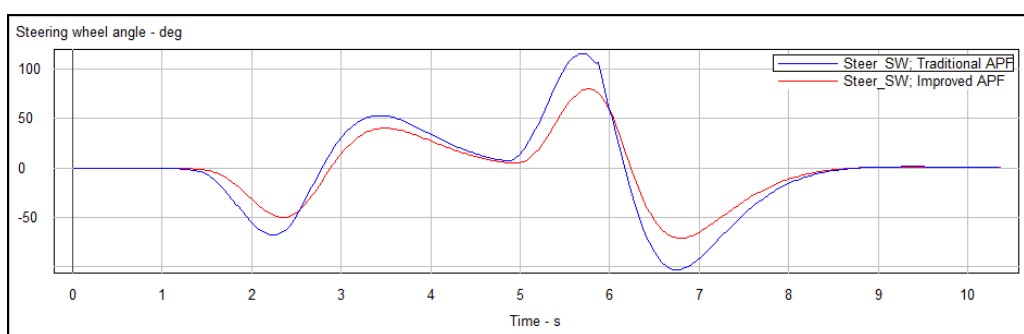

**Figure 26.** Steering wheel angles for expected vehicle speed of 10 m/s in dynamic obstacles driving environment.

Figure 27 compared the planned trajectory and the actual trajectory for expected vehicle speed of 15 m/s. The Euclidean errors of the planned trajectory and the actual trajectory for expected vehicle speed of 15 m/s are shown in Figure 28. The variation curves of steering wheel angle for expected vehicle speed of 15 m/s in dynamic obstacles driving environment are shown in Figure 29.

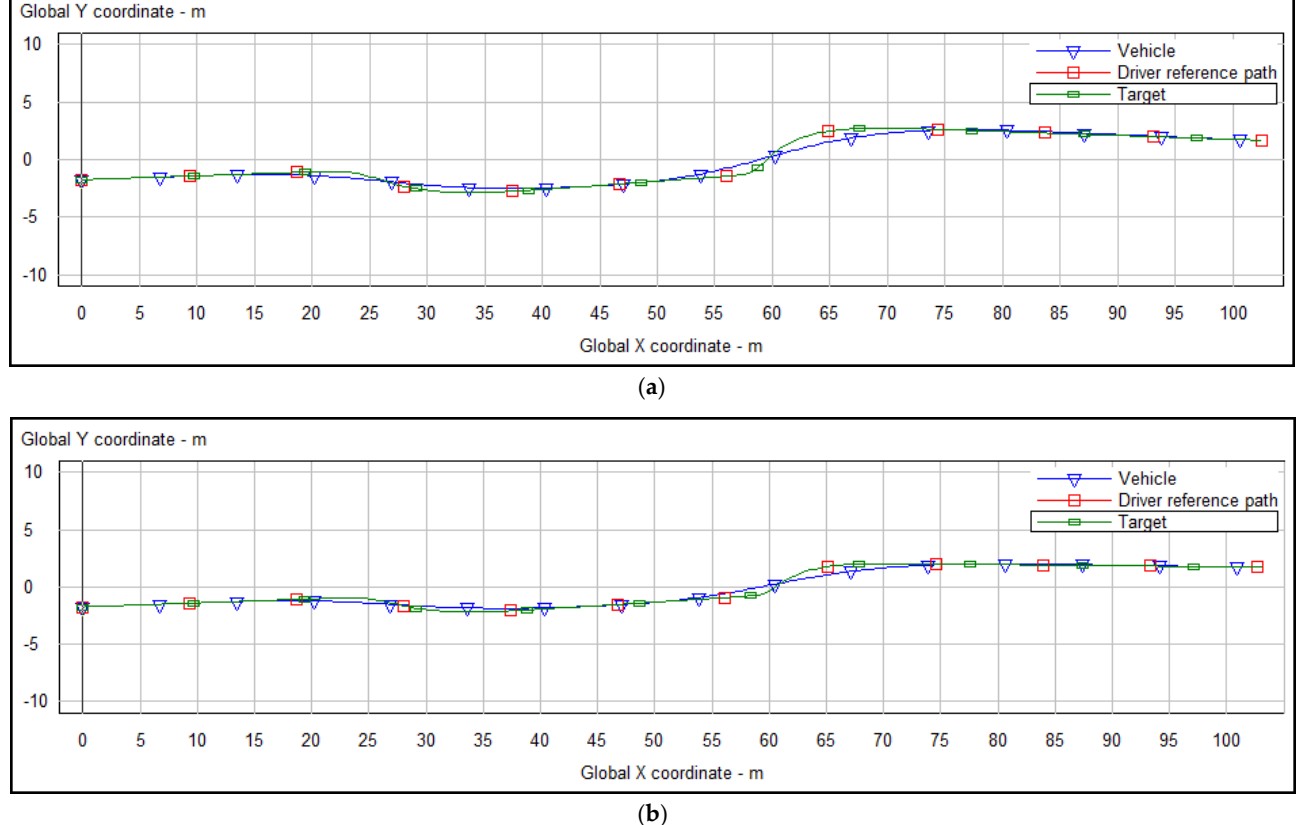

(a)

(b)

**Figure 27.** Comparison of planned trajectory and actual trajectory for expected vehicle speed of 15 m/s in dynamic obstacles driving environment. (**a**) Planned trajectory by traditional APF and actual trajectory of vehicle; (**b**) planned trajectory by improved APF and actual trajectory of vehicle.

As can be seen in figure about Euclidean errors of trajectories planned, by using the same Carsim tracking model, the tracking Euclidean error of the trajectory planned by the improved APF is smaller, which indicates that the trajectory planned by the improved APF method is easier to be tracked. Due to the better tracked trajectory obtained, the steering wheel angle of the vehicle is smaller when tracking the trajectory planned by the improved APF. It indicates that the intelligent vehicle is easier to be maneuvered.

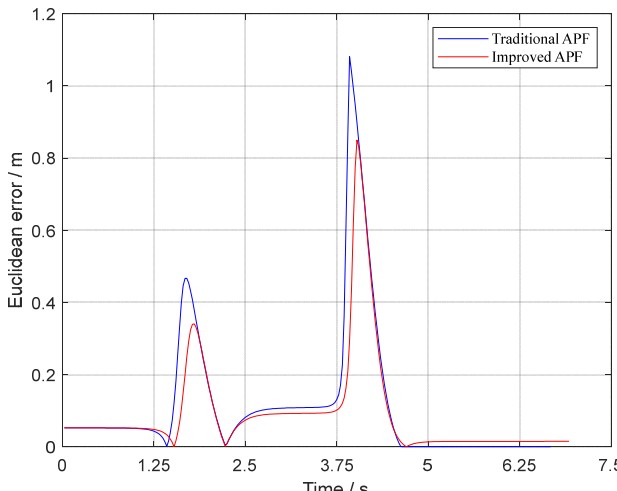

**Figure 28.** Euclidean errors of trajectories planned for expected vehicle speed of 15 m/s in dynamic obstacles driving environment.

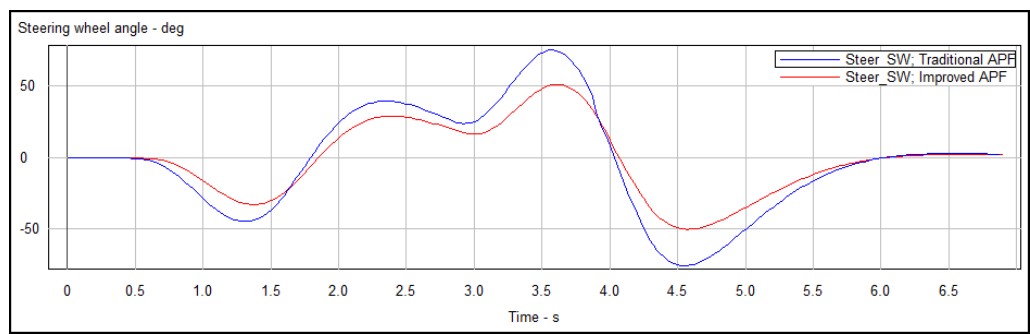

**Figure 29.** Steering wheel angles for expected vehicle speed of 15 m/s in dynamic obstacles driving environment.

## 6. Conclusions

(1) This paper introduces the principle of the traditional APF method and its advantages and shortcomings, solving the problem of excessive initial attractive force and intelligent vehicle cannot reach the target by improving the potential field functions. At the same time, establishing the road boundary potential field combined with the actual application scenario.

(2) A strategy of jump out of local minima based on smaller steering angles has been proposed, solving the problem of local minima that the traditional APF method tends to fall into by finding smaller steering angles and determining the appropriate jump out step length in the steering angle range of the vehicle.

(3) The improved APF method can not only jump out local minima but also plan smooth trajectories by simulation in Matlab. By comparing the magnitude of curvature and tracking the planned trajectories in Carsim platform, the reduction of Euclidean error and steering wheel angle proved that the trajectories planned by the improved APF method are easier to be tracked.

However, there are two drawbacks in the research of this paper:

1. The driving environment designed in this paper is urban road and the general vehicle speed limit range is 30–60 km/h in the city. 10 m/s (36 km/h) and 15 m/s (54 km/h) are the common speeds in the speed limit range, so they are used as the simulation speed in this paper. Furthermore, obstacle avoidance of high-speed vehicles is a complex motion planning involving braking and steering. If the speed of the vehicle is too high, the actual trajectory will have a large deviation from the planned trajectory

due to the inertia of the vehicle, which may directly cause the vehicle to have a lateral collision with the obstacle and lead to the failure of the path planning. In this paper, under the assumption of uniform vehicle motion, the vehicle trajectory planning under high-speed motion is not considered.

2. The APF model in this paper does not consider the differences of obstacle avoidance trajectories of different vehicle types in the actual road environment, and only considers the obstacle avoidance scenarios of flat and straight roads, which is a relatively single scene.

In view of the above deficiencies, the subsequent research will keep improve the APF model and enhance the adaptability of the model. Hence, future work should be devoted to establishing the APF model under high-speed motion. In addition, an avenue for our future work would be to research the interaction between the subject vehicle and the obstacle vehicle with different parameters in the obstacle avoidance process.

**Author Contributions:** Conceptualization, J.T.; methodology, J.T.; software, J.T.; validation, J.T.; formal analysis, B.L. and S.B.; investigation, J.T.; resources, J.T., B.L. and S.B.; data curation, J.T.; writing—original draft preparation, J.T., H.H. and Z.Q.; writing—review and editing, J.T., S.B., B.L., D.Z., X.Z. and H.T.; visualization, J.T.; supervision, B.L. and S.B.; project administration, B.L. and S.B.; funding acquisition, not applicable. All authors have read and agreed to the published version of the manuscript.

**Funding:** This research was funded by the Natural Science Foundation of the Jiangsu Higher Education of China under grant number 21KJA580001, the National Natural Science Foundation of China under grant number 52172367, and the National Natural Science Foundation of China under grant number 52105260. The APC was funded by 52172367.

**Institutional Review Board Statement:** Not applicable.

**Informed Consent Statement:** Not applicable.

**Data Availability Statement:** Not applicable.

**Conflicts of Interest:** The authors declare no conflict of interest.

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
