# Peer review of "Research on Active Obstacle Avoidance of Intelligent Vehicles Based on Improved Artificial Potential Field Method"

_wevj, doi:10.3390/wevj13060097_

Round 1

Reviewer 1 Report

The paper entitled “Research on Active Obstacle Avoidance of Intelligent Vehicles Based on Improved Artificial Potential Field Method” studies the problem of autonomous obstacle avoidance in intelligent vehicles. Simulation results are provided to validate the proposed approach.

However, I have some concerns as follows:

Technical aspect:

  • The literature review section at the beginning needs improvement to better represent the state-of-the-art regarding the planning techniques. I would recommend the respected authors to study the following works (and the references therein) regarding the planning algorithms (classical and state-of-the-art) and address them appropriately:

LaValle, S. M. (2006). Planning algorithms. Cambridge university press.

Vega-Brown, W., & Roy, N. (2020). Asymptotically optimal planning under piecewise-analytic constraints. In Algorithmic Foundations of Robotics XII (pp. 528-543). Springer, Cham.

Hasankhani, A., Ondes, E. B., Tang, Y., Sultan, C., & VanZwieten, J. (2021). Integrated Path Planning and Tracking Control of Marine Current Turbine in Uncertain Ocean Environments. arXiv preprint arXiv:2110.07105.

Jafari, M., Xu, H., & Carrillo, L. R. G. (2018, November). Brain emotional learning-based path planning and intelligent control co-design for unmanned aerial vehicle in presence of system uncertainties and dynamic environment. In 2018 IEEE Symposium Series on Computational Intelligence (SSCI) (pp. 1435-1440). IEEE.

Chen, Y., Liang, J., Wang, Y., Pan, Q., Tan, J., & Mao, J. (2020). Autonomous mobile robot path planning in unknown dynamic environments using neural dynamics. Soft Computing, 1-17.

Tatehara, T., Nagahama, A., & Wada, T. (2022). Online Maneuver Learning and its Real-Time Application to Automated Driving System for Obstacles Avoidance. IEEE Transactions on Intelligent Vehicles.

Nakrani, N. M., & Joshi, M. M. (2022). A human-like decision intelligence for obstacle avoidance in autonomous vehicle parking. Applied Intelligence, 52(4), 3728-3747.

González, D., Pérez, J., Milanés, V., & Nashashibi, F. (2015). A review of motion planning techniques for automated vehicles. IEEE Transactions on Intelligent Transportation Systems, 17(4), 1135-1145.

Zhang, Z., Wang, C., Zhao, W., & Feng, J. (2022). Longitudinal and lateral collision avoidance control strategy for intelligent vehicles. Proceedings of the Institution of Mechanical Engineers, Part D: Journal of Automobile Engineering, 09544070211024048.

  • I highly recommend the respected authors to study the following work regarding the planning techniques:

LaValle, S. M. (2006). Planning algorithms. Cambridge university press.

  • The respected authors referred to ref. [8] as Artificial Potential Field method proposed by Khatib, however, ref. [8] in this paper is “Ouyang, X.; Yang, S. Obstacle Avoidance Path Planning of Mobile Robots Based on Potential Grid Method [J]. Control Engineering of China, ,2014,21(01):134-137.”.

I suggest the respected authors to properly refer to the articles in the literature.

  • The respected authors mentioned “RRT (Rapid-exploration Random Tree)”, however, I believe RRT stands for “Rapidly-exploring random trees” and was introduced in “LaValle, S. M. (1998). Rapidly-exploring random trees: A new tool for path planning.”

I suggest the respected authors to use the correct terms and to properly refer to this original article.

  • From practical point of view, dynamic of the vehicles is very important in designing the feasible path. I highly recommend the respected authors to address this in their manuscript.

Presentation aspect:

  • The paper needs improvement in presentation. For example, there are too many typos, formatting errors, and it seems that the authors did not review the paper themselves before submitting it. For example, there are several instances of “[Error! Reference source not found.]” in the current version of the manuscript.

Reviewer 2 Report

The subject of the article and the research conducted is very interesting, however, the article contains several shortcomings and errors, indicated below:

  1. The Introduction section should be slightly changed - there is no introductory information on the subject of intelligent vehicles and an explanation why the issue of route planning is so important.
  2. In lines 36-38, the authors mention path planning methods: Dijkstra algorithm, A * algorithm, RRT (Rapid-exploration Random Tree) algorithm, grid method, APF (Artificial Potential Field), while further in the text, only the APF method is described in detail. In my opinion, it is also worth describing the other methods in 2-3 sentences and, above all, explaining the differences between them, as well as why the authors decided to use this particular method in their research.
  3. Authors should avoid too long paragraphs - e.g. a paragraph contained on lines 39-75 - individual areas should be separated from each other in a meaningful way to make the text more readable.
  4. At the end of the Introduction section, the aim of the article should be clearly defined.
  5. Is section 2 a Literature Review? Because only the APF method is described in it, and where is the other relevant information on the subject of the article?
  6. Figures 1 and 2 do not have the references provided - are these figures from the authors' own research? Or are they copied from some references?
  7. Where do equations 1-6 come from? There is no given reference.
  8. In Subsection 2.2, the authors mention shortcomings and improvement, but of what? By default, of course, it is known that it is the APF method, but it should either be given specifically in the title or at the beginning of this subsection there should be 2-3 introductory sentences explaining what the following points are about and what they will contain.
  9. Based on what literature references was subsection 2.2 written?
  10. Only the results of the conducted research are presented in the article. However, there is no discussion of the results obtained.
  11. The conclusions are not very extensive.
  12. Not all references are prepared in accordance with the editorial requirements. Besides, very few references are up-to-date - many of them come from over a dozen years ago.
  13. In some places there is no literature source given - instead, there is the statement: "[Error! Reference source not found." - please correct it (e.g. line 49, 52, etc.).
  14. Please also make sure that the entire article is correct and also rewrite some sentences, e.g. "Ultimately, the combined forces control the movement of the vehicle. As shown in the Figure 2" - lines 125-126.

Authors should also pay attention to editorial aspects - e.g. there are no spaces in several places.

Reviewer 3 Report

The paper follows a field theoretical approach using artifical potentials, to supply intelligent vehicles with information on approaching obstacles on the road.  

While the idea in principle sounds interesting, the merit of the study is unclear. On top of that, the paper has plenty of editorial and language errors.

1) Figures 17-20 show the stearing wheel angle versus step. In the current version, the figures do not tell the reader anything about quality of the estimates. The ground-truth must be added to the curves; and the (eucledean) error has to be shown for both, traditional and improved APFs in separate plots.  Moreover, it is better to label the x-axis with time in [s] rather than steps (undefined).  

2) Multiple references are undefined such as in line 49, 52, 62, and 66

3) There are plenty of language errors. The worst ones are: i) line 17: What is a  "jump-out step". This phrase does not exist in English. Probably, the author meant "opt-out step"? ii) What is Carsim platform in line 22?

4) Fig. 17-20 only focus on 10m/s and 15m/s. What happens when the velocity of the vehicle is ranged from 0 to 45m/s  (highway) -- or where is the limit of operation? What is the performance of the algorithm as function of speed?   

5) What are the limits and drawbacks of the presented method? A discussion on this issues is completely missing and hence, has to be added - at least in the conclusions.

6) Relevant references are missing:

Bing Lu et al, 2020, DOI: 10.1109/ACCESS.2020.3044909
Hongyu et al, 2018, DOI: 10.1016/j.ifacol.2018.10.095
Wang et al, 2022, DOI: 10.3390/act11020052

Round 2

Reviewer 1 Report

N/A

Author Response

We thank the reviewers for their constructive criticism that has helped us to improve the manuscript. We have revised the manuscript again and hope to meet the publication requirements.

Reviewer 2 Report

The authors addressed the comments and suggestions carefully, and made significant changes to improve the manuscript. The article may be published in its present form.

Author Response

We thank the reviewers for their constructive criticism that has helped us to improve the manuscript. 

Reviewer 3 Report

Quality of the mansucript has improved. However, the following (mandatory) issues are still open:

1)  For the missing plots showing ground truth, it is fine to add the true trajectory (in readable format) as suggested by the author:

"I'm sorry I couldn't understand the "the ground-truth " you mentioned. Is it a comparison of the vehicle's planned trajectory and its actual movement? I prepared a comparison diagram between the trajectory planned by the traditional APF method and the actual trajectory of the vehicle at 15m/s. If my understanding is correct, I will add it in the next revision; if my understanding is wrong, please point out my mistake. Thanks again to the respected reviewer."

2) For the limits of the proposed approach, the authors replied "In response to the reviewer's question about what happens when the velocity of the vehicle is ranged from 0 to 45m/s (highway), the intelligent vehicle speed below 80km/h has better experimental results after our test. When the vehicle speed is higher than 80km/h, the vehicle may have a large tracking error.( because we used Carsim's default tracking model). "

It should be clearly stated in the conclusions what to expect at higher speeds and why the authors did not look at this scenario at the moment.

Round 3

Reviewer 3 Report

The authors have considered the reviewer's concerns.